# Leakage Benchmarking for Universal Gate Sets

**DOI:** 10.3390/e26010071

**Published:** 2024-01-13

**Authors:** Bujiao Wu, Xiaoyang Wang, Xiao Yuan, Cupjin Huang, Jianxin Chen

**Affiliations:** 1Center on Frontiers of Computing Studies, Peking University, Beijing 100871, China; bujiaowu@gmail.com (B.W.); zzwxy@pku.edu.cn (X.W.); xiaoyuan@pku.edu.cn (X.Y.); 2School of Computer Science, Peking University, Beijing 100871, China; 3School of Physics, Peking University, Beijing 100871, China; 4Alibaba Quantum Laboratory, Alibaba Group USA, Bellevue, WA 98004, USA; jianxinchen@acm.org

**Keywords:** quantum computing, randomized benchmarking, leakage error, quantum gates

## Abstract

Errors are common issues in quantum computing platforms, among which leakage is one of the most-challenging to address. This is because leakage, i.e., the loss of information stored in the computational subspace to undesired subspaces in a larger Hilbert space, is more difficult to detect and correct than errors that preserve the computational subspace. As a result, leakage presents a significant obstacle to the development of fault-tolerant quantum computation. In this paper, we propose an efficient and accurate benchmarking framework called *leakage randomized benchmarking* (LRB), for measuring leakage rates on multi-qubit quantum systems. Our approach is more insensitive to state preparation and measurement (SPAM) noise than existing leakage benchmarking protocols, requires fewer assumptions about the gate set itself, and can be used to benchmark multi-qubit leakages, which has not been achieved previously. We also extended the LRB protocol to an interleaved variant called interleaved LRB (iLRB), which can benchmark the average leakage rate of generic *n*-site quantum gates with reasonable noise assumptions. We demonstrate the iLRB protocol on benchmarking generic two-qubit gates realized using flux tuning and analyzed the behavior of iLRB under corresponding leakage models. Our numerical experiments showed good agreement with the theoretical estimations, indicating the feasibility of both the LRB and iLRB protocols.

## 1. Introduction

Quantum computation maps information processing into the manipulation of (typically microscopic) physical systems governed by quantum mechanics. Although quantum computation holds the promise to solve problems that are believed to be classically intractable, practical quantum computation suffers from various noise sources, ranging from fabrication defects and control inaccuracies to fluctuations in external physical environments. Such noise greatly hinders the practicability of quantum computation on unprotected, bare physical qubits beyond proof-of-concept demonstrations.

While any kind of error is unwanted and would possibly affect the quality of the computation processes, there is a significant difference between the harmfulness of different types of errors. The most “benign” error happens locally and independently on single qubits; such errors can, in principle, be compressed arbitrarily with quantum error correction under reasonable assumptions on the error rates [1,2]. More-malicious errors might introduce time correlations (e.g., non-Markovian errors) or space correlations (e.g., crosstalk) and are more challenging to mitigate. Of particular interest is the *leakage error*, where a piece of quantum information escapes from a confined, finite-dimensional Hilbert space used for computation, called the *computational subspace*, to a *leaked subspace* of a larger Hilbert space. Such escaped information might undergo arbitrary and uncontrolled processes and is harder to detect, let alone correct. More seriously, typical frameworks of quantum error correction only deal with errors happening within the computational subspace and are either unable to be applied or scale poorly with the leakage error. It is, thus, of great importance to be able to detect, correct, or even suppress leakage errors in order to conduct large-scale quantum computation.

This paper focuses on estimating the leakage error rate associated with a given quantum processor, preferably efficiently and accurately. This task is part of a process usually referred to as *benchmarking*, providing an estimate of certain characteristics of a piece of the quantum device before proceeding with subsequent actions. In the context of leakage benchmarking, the information can be used as a criterion to accept or abort a newly fabricated quantum processor or as feedback information on leakage-suppressing gate schemes.

Given the diverse nature of errors occurring in quantum computation, many different benchmarking schemes have been proposed over the years. A large class of benchmarking schemes, collectively called *randomized benchmarking* (RB), extracts error information from the fit result of multiple experiments with different lengths [3,4,5,6,7,8,9,10,11,12,13,14]. Compared to tomography-based methods or direct fidelity estimation [15,16], RB schemes are typically more gate-efficient, and the fitting results are typically insensitive to state preparation and measurement (SPAM) errors, making them ideal candidates for benchmarking gate errors. These protocols have been successfully implemented in many quantum experiments [17,18,19,20,21].

The first theoretical framework for RB-based leakage benchmarking was given by Wallman et al. [22]. Without any prior assumption on the SPAM noise, this protocol was able to provide an estimate for the sum of the leakage rate and the *seepage rate*, i.e., the rate information in the leaked subspace comes back to the computational subspace. Refs. [8,23] later gave a detailed analysis of the protocol and illustrated this framework with several examples relevant to superconducting devices. The authors were also able to differentiate the leakage from the seepage with reasonable assumptions on the SPAM noise. Based on these protocols, several experimental characterizations of single-qubit leakage noise have been proposed in superconducting quantum devices [24,25], quantum dots [26], and trapped ions [27].

There are two major limitations to the existing protocols [8,11,22,23]. First, all protocols require that the quantum gates act nontrivially on the leakage subspaces, in order to eliminate non-Markovian behavior originating from residual information stored in the leakage subspace. As most practical gate schemes only focus on their actions on computational subspaces rather than the leakage subspaces, leakage benchmarking schemes built upon them typically do not work in general, multi-qubit quantum systems. Second, most existing protocols can only estimate the sum of the leakage rate and the seepage rate without prior knowledge of SPAM noise, and the SPAM information is required if we need to obtain the leakage and seepage rates separately. As there is typically only one set of state preparation and measurements within one run of benchmarking, the SPAM errors do not become amplified and cannot be measured accurately [28]. Such inaccuracy would further affect the accuracy of gate leakage rate estimation. A natural question arises: *How can one characterize the leakage rate of a multi-qubit system without operating the leakage subspace while maintaining robustness to SPAM noise?*

In this paper, we propose a leakage benchmarking scheme based on RB, dedicated to benchmark leakage rates on multi-qubit systems. Compared to existing protocols requiring the leakage subspace to be fully twirled, our scheme only requires having access to the Pauli group with gate-independent, time-independent, and Markovian noise. Assuming each qubit has only one-dimensional leakage space, such a gate set does not twirl the leakage subspace as a whole, but instead twirls each invariant subspace of the Pauli group individually. This allows us to formulate the LRB process as a classical Markovian process between different invariant subspaces, which can be described by a Markovian *Q*-matrix [29]. The leakage and seepage rates of the system can then be estimated by leveraging the spectral property of the Markovian process, which can, in turn, be estimated similarly to RB protocols on the computational subspace.

The *Q*-matrix has a dimension exponential with respect to the number of qubits in general, and thus, the spectral property is hard to measure using LRB experiments. To further simplify the problem, we studied the spectrum of the *Q*-matrix in two physically motivated scenarios: The first model, named *leakage damping noise*, assumes that leakage happens at most one qubit and leakage does not “hop” from one qubit to another, which is the generalization of amplitude damping noise [30] in the computational subspace; the second model assumes that each qubit undergoes an independent leakage process. In both cases, the spectral property of the *Q*-matrix can be significantly simplified and easier for data analysis. We also show how to calculate the corresponding average leakage rates on the above two noise scenarios of the proposed LRB protocol. As an illustration of the leakage damping noise model, we found that the noise model of commonly used two-qubit gates such as iSWAP, SQiSW, and CZ gates belongs to this form.

Building upon the foundation of leakage randomized benchmarking (LRB) protocols, we delve deeper into the study of leakage benchmarking for specific multi-qubit gates, which is a crucial aspect of quantum hardware development. To this end, we propose an interleaved variant of the LRB (iLRB) protocol that allows for the benchmarking of individual gates, rather than a set of gates. We show that the leakage rate can be extracted in general for arbitrary target gates with access to noiseless Pauli gates and performed a more-careful analysis when Pauli gates were implemented as noisy. In addition, we show that the leakage rate of the target gate can still be extracted under certain physically motivated assumptions that inherently apply to flux-tuning gates in superconducting quantum computation. To demonstrate the applicability of the iLRB protocol, we applied it to the case of flux tunable superconducting quantum devices [31], constructed its noise model, and benchmarked the leakage rate of the iSWAP gate.

This paper targets both theorists and experimentalists, as it seeks to establish an experiment-friendly leakage benchmarking scheme. We offer a thorough theoretical analysis for multi-qubit scenarios, as well as a numerical verification of the average leakage rate for the iSWAP gate. This was achieved by extracting the noise model of the iSWAP gate from its Hamiltonian evolution.

In Section 2, we introduce the fundamental concepts and notations. Section 3 presents our LRB protocol and analyzes the calculation of the average leakage rate using this method. In Section 4, we provide a detailed examination of the average leakage rate under two leakage models: single-site leakage and no crosstalk. Section 5 proposes the iLRB protocol for any target gate that commutes with the noise channel, focusing on a special leakage damping noise. In Section 6, we numerically validate the LRB and iLRB protocols. Additionally, we introduce the leakage damping noise model for iSWAP/SQiSW/CZ gates in flux-tunable superconducting quantum devices, based on their Hamiltonian evolution. We also tested the iLRB protocols numerically using the noise model of the iSWAP gate. Finally, Section 7 concludes the paper with a discussion of our work and suggestions for future research directions.

## 2. Notations

In order to characterize leakage, we assume that the quantum states lie in a Hilbert space H with finite dimension *d* that decomposes into a *computational* and a *leakage* subspace, denoted as $H_{c}$ and $H_{l}$ respectively. Let $d_{c}:=\dim\left(H_{c}\right)$ and $d_{l}:=\dim\left(H_{l}\right)=d-d_{c}$ be the dimensions of $H_{c}$ and $H_{l}$. Unless explicitly specified, we assume throughout the paper that a single qubit (site) lies in a three-dimensional Hilbert space with basis {|0〉,|1〉,|2〉}, where the computational subspace is spanned by {|0〉,|1〉} and the leakage subspace by {|2〉}. In other words, higher-level excitations of a qubit can be ignored. We call such a system *a single qubit with leakage*.

A composite system of *n* qubits with leakage lies in a Hilbert space $H={⨂}_{k=1}^{n}\left(H_{c_{k}}⊕H_{l_{k}}\right)$, where $H_{c_{k}}$ ($H_{l_{k}}$) represents the computational (leakage) subspace of the qubit *k*. We define the computational subspace of H be where no qubits leaks, that is, $H_{c}={⨂}_{k=1}^{n}H_{c_{k}}$. Hence $d=3^{n}$, and $d_{c}=2^{n}$. The projector on the computational subspace $\Pi_{c}={⊗}_{k=1}^{n}\Pi_{c_{k}}$ is a tensor product where $\Pi_{c_{k}}$ is the projector onto the computational subspace on the *k*-th qubit. Note that the projector onto the leakage subspace on the *k*-th qubit is $\Pi_{l_{k}}=\left|2\rangle \langle 2\right|$ and the projector onto the leakage subspace $\Pi_{l}:=I-\Pi_{c}\neq {⊗}_{k=1}^{n}\Pi_{l_{k}}$, where I is the identity operator on H. For each $i=\left(i_{1},i_{2},\cdots ,i_{n}\right)\in {\left\{c,l\right\}}^{n}$, we define $H_{i}:={⨂}_{k=1}^{n}H_{i_{k}}$ to be the subspace where qubit *k* is leaked if and only if $i_{k}=l$. The corresponding projector onto $H_{i}$ is $\Pi_{i}:={⊗}_{k=1}^{n}\Pi_{{\left(i_{k}\right)}_{k}}$. Note that $H={⨁}_{i\in {\left\{c,l\right\}}^{n}}H_{i},H_{c}=H_{c^{n}}$, and $H_{l}={⨁}_{i\neq c^{n}}H_{i}$. For each Hilbert space $H_{i}$, denote ${\tilde{\Pi }}_{i}:=\Pi_{i}/\dim\left(H_{i}\right)$ the trace-normalized projector associated to the projector $\Pi_{i}$.

We assume the noise of interest to be Markovian and time-independent throughout this paper. Given an ideal unitary $U\in U(d_{c})$, we denote $U\left(\cdot \right):=\left(\Pi_{l}⊕U\right)\cdot \left(\Pi_{l}⊕U^{†}\right)$ as the corresponding ideal unitary channel acting on the whole space. Given a completely positive trace-preserving (CPTP) channel $\hat{U}$ characterizing the noisy implementation of U, we further denote $\Lambda :=U^{†}\circ \hat{U}$ as the noise information of U accounting leakage. Note that $\hat{U}=U\circ \Lambda$ as U is a unitary channel. The average leakage and seepage rates of a channel Λ are defined as [23]
(1)$$\begin{array}{c} L_{\mathrm{ave}}\left(\Lambda \right)=\mathrm{Tr}\left(\Pi_{l}\Lambda \left({\tilde{\Pi }}_{c}\right)\right); \end{array}$$
(2)$$\begin{array}{c} S_{\mathrm{ave}}\left(\Lambda \right)=\mathrm{Tr}\left(\Pi_{c}\Lambda \left({\tilde{\Pi }}_{l}\right)\right). \end{array}$$
We often write $L_{\mathrm{ave}}$ and $S_{\mathrm{ave}}$ when the noise channel Λ being referred to is unambiguous. Unless explicitly specified, we use the term “leakage noise” to represent both leakage and seepage errors.

The Pauli group with phase P<U(2) is defined as $\pm \left\{1,i\right\}\times \left\{I,X,Y,Z\right\}$, where X,Y,Z are 2×2 Pauli-X/Y/Z matrices respectively. Let $P_{n}:=P^{\times n}<U{\left(2\right)}^{\times n}$. For an element $P={⨂}_{i}P_{i}\in P_{n}$, its corresponding ideal unitary channel in the full space is defined as $P:={⨂}_{i}P_{i}$. For sake of simplicity, we identify the element *P* with its corresponding ideal channel P, and use $\hat{P}$ as a shorthand for the corresponding noisy implementation P∘Λ.

Inspired by the Pauli-transfer matrix (PTM) representation [32], here we define the *condensed-operator representation* |·)) of linear operators as the Liouville representation [33] with respect to the orthonormal operator basis $I={\left\{\Pi_{i}/\sqrt{\dim(H_{i})}\right\}}_{i\in {\left\{c,l\right\}}^{n}}$. The basis is not complete in the sense that it does not span L(H); for a linear operator ρ not lying in the span of I, |ρ)) is understood as the projection of ρ onto the span of I followed by the vectorization, that is,
$$\left|\rho )):=\right|\bar{P}\left(\rho \right))),$$
where $\bar{P}:L\left(H\right)\to \mathrm{span}\left(I\right);\bar{P}\left(\rho \right):=\sum_{i}\mathrm{Tr}\left(\Pi_{i}\rho \right){\tilde{\Pi }}_{i}$ is the *twirling projector* from L(H) to span(I).

For sake of clarity, in the following, we represent the condensed operator representations under the basis $\left\{|{\tilde{\Pi }}_{i}\right){)\}}_{i}$, and the adjoints under the basis $\{({\left(\Pi_{i}|\right\}}_{i}$. Note that $\left(\left(\Pi_{i}|{\tilde{\Pi }}_{j}\right)\right)=\delta_{ij}$. Under such basis choice, for a generic linear operator A∈L(H), we have
$$\left|A)\right)=\sum_{i}\mathrm{Tr}\left(\Pi_{i}A\right)|{\tilde{\Pi }}_{i}))\;\mathrm{and}\;((A|=\sum_{i}\mathrm{Tr}\left({\tilde{\Pi }}_{i}A^{†}\right)((\Pi_{i}|.$$
For a superoperator Λ, the corresponding condensed operator representation is then
(3)$$Q_{\Lambda }:=\sum_{i,j}\mathrm{Tr}\left(\Pi_{i}\Lambda \left({\tilde{\Pi }}_{j}\right)\right)|{\tilde{\Pi }}_{i}))((\Pi_{j}|.$$

Since I does not form a complete basis, compositions of condensed operator representations do not directly translate to compositions of the corresponding linear operators; rather they translate to compositions of the twirled versions of the corresponding linear operators through the twirling projector $\bar{P}$. More specifically, we have
(4)$$\begin{array}{cc} Q_{\Lambda_{1}}Q_{\Lambda_{2}} & =Q_{\Lambda_{1}\circ \bar{P}\circ \Lambda_{2}}, \end{array}$$
(5)$$\begin{array}{cc} Q_{\Lambda }\left|\rho )\right) & =|\left(\Lambda \circ \bar{P}\right)\left(\rho \right))), \end{array}$$
(6)$$\begin{array}{cc} \left((M\right|Q_{\Lambda }\left|\rho )\right) & =\mathrm{Tr}\left(M\cdot \left(\bar{P}\circ \Lambda \circ \bar{P}\right)\left(\rho \right)\right). \end{array}$$
We denote [n]:={1,…,n} throughout the paper.

## 3. Leakage Randomized Benchmarking Protocol

Here we present a leakage randomized benchmarking protocol that does not require actions on the leakage subspace or assumptions about SPAM errors. Our protocol is based on the assumption that the noise, represented by the operator Λ, is Markovian, time-independent, and gate independent. We further assume we have access to a noisy measurement operator ${\hat{\Pi }}_{c}$ close to the projector to the computational subspace $\Pi_{c}$.

(1)Given a sequence length *m*, sample a sequence of *m* Paulis $P_{1},\ldots ,P_{m}$ from $P_{n}$ uniformly i.i.d., and perform them sequentially to a fixed (noisy) initial state ${\hat{\rho }}_{0}$, obtaining ${\hat{P}}_{m}\circ \cdots \circ {\hat{P}}_{1}\left({\hat{\rho }}_{0}\right)$. Measure the output state under ${\hat{\Pi }}_{c}$ and estimate the probability $p_{\Pi_{c}}\left(P_{1},\ldots ,P_{m}\right)=\mathrm{Tr}\left({\hat{\Pi }}_{c}{\hat{P}}_{m}\circ \cdots \circ {\hat{P}}_{1}\left({\hat{\rho }}_{0}\right)\right)$ through repeated experiments.(2)Repeat Step (1) multiple times to estimate $p_{\Pi_{c}}\left(m\right)$, the expectation of $p_{\Pi_{c}}\left({\hat{P}}_{1},\ldots ,{\hat{P}}_{m}\right)$ under random choices of $P_{1},\ldots ,P_{m}$ from $P_{n}$.(3)Repeat Step (2) for different *m*, and fit $\left\{\left(m,p_{\Pi_{c}}\left(m\right)\right)\right\}$ to a multi exponential decay curve $p_{\Pi_{c}}\left(m\right)=\sum_{i}A_{i}\cdot \lambda_{i}^{m}$.

The average leakage rate $L_{\mathrm{ave}}$ and seepage rate $S_{\mathrm{ave}}$ are estimated with the fitted exponents $\lambda_{i}$. The number of exponents for $p_{\Pi_{c}}\left(m\right)$ depends on the specific noise model of Λ. In the following, we will show the explicit representation of $E\left[p_{\Pi_{c}}\left(m\right)\right]$ and $\lambda_{i}$.

The Pauli group $P_{n}$ can twirl any quantum state in computational subspace to the maximum mixed state [34,35], i.e., $\frac{1}{|P_{n}|}\sum_{P_{c}\in P_{n}}P_{c}\left(\rho_{c}\right)=\mathrm{Tr}\left(\rho_{c}\right){\tilde{\Pi }}_{c}$, where $\rho_{c}\in L\left(H_{c}\right)$ is a quantum state in computational subspace. Here we expand the twirling of a Pauli group from computational subspace to the entire Hilbert space, as shown in Lemma 1.

**Lemma** **1.**
*Let $\bar{P}$ be the twirling projector such that $\bar{P}\left(\rho \right)=\sum_{i}\mathrm{Tr}\left(\Pi_{i}\rho \right){\tilde{\Pi }}_{i}$ for any quantum state ρ. Then it can be equivalently represented as the expectation of all the Pauli channels,*

(7)
$$\bar{P}=\frac{1}{|P_{n}|}\sum_{P\in P_{n}}P.$$



Lemma 1 can be obtained from the twirling properties of Pauli group $P_{n}$ in the computational subspace. We postpone the proof of Lemma 1 into Appendix A. With Lemma 1, we can construct the connections of $L_{\mathrm{ave}},S_{\mathrm{ave}}$ and the multi-exponential decay curve $p_{\Pi_{c}}\left(m\right)$, as shown in the following theorem.

**Theorem** **1.***Given Pauli group $P_{n}$ with gate-independent leakage error channel *Λ, *the average output probability in LRB protocol $p_{\Pi_{c}}\left(m\right)=(({\hat{\Pi }}_{c}|Q^{m-1}|{\tilde{\rho }}_{0}))$, where $Q:=Q_{\Lambda }$ is the condensed-operator representation of *Λ *and ${\tilde{\rho }}_{0}$ is some noisy state determined by the input state ${\hat{\rho }}_{0}$. The average leakage rate equals $L_{\mathrm{ave}}=1-Q_{c^{n},c^{n}}$ and the average seepage rate equals $S_{\mathrm{ave}}=\frac{1}{3^{n}-2^{n}}\sum_{i\neq c^{n}}\dim\left(\Pi_{i}\right)Q_{c^{n},i}$.*

**Proof.** Let $P_{1},\ldots ,P_{m}$ be the ideal gate elements sampled from $P_{n}$. Then the expectation of the probability for measuring computational basis equals
(8)$$\begin{array}{ccc} \; & p_{\Pi_{c}}\left(m\right) & \;\;=\frac{1}{|P_{n}{|}^{m}}\sum_{P_{1},\ldots ,P_{m}\in P_{n}}\mathrm{Tr}\left({\hat{\Pi }}_{c}{\hat{P}}_{m}\circ \cdots \circ {\hat{P}}_{1}\left({\hat{\rho }}_{0}\right)\right) \end{array}$$
(9)$$\begin{array}{cc} \; & =\mathrm{Tr}\left({\hat{\Pi }}_{c}{\left(\frac{1}{|P_{n}|}\sum_{P\in P_{n}}P\circ \Lambda \right)}^{m}\left({\hat{\rho }}_{0}\right)\right) \end{array}$$
(10)$$\begin{array}{cc} \; & =\mathrm{Tr}\left({\hat{\Pi }}_{c}{\left(\bar{P}\circ \Lambda \right)}^{m}\left({\hat{\rho }}_{0}\right)\right) \end{array}$$
(11)$$\begin{array}{cc} \; & =\mathrm{Tr}\left({\hat{\Pi }}_{c}{\left(\bar{P}\circ \Lambda \right)}^{m-1}\bar{P}\circ \Lambda ({\hat{\rho }}_{0}\right)) \end{array}$$
(12)$$\begin{array}{cc} \; & \;\;=\mathrm{Tr}\left({\hat{\Pi }}_{c}\bar{P}\circ \left(\Lambda \circ {\left(\bar{P}\circ \Lambda \right)}^{m-2}\right)\circ \bar{P}\circ \Lambda ({\hat{\rho }}_{0})\right) \end{array}$$
(13)$$\begin{array}{cc} \; & =(({\hat{\Pi }}_{c}|Q_{\Lambda \circ {\left(\bar{P}\circ \Lambda \right)}^{m-2}}|{\tilde{\rho }}_{0})) \end{array}$$
(14)$$\begin{array}{cc} \; & =(({\hat{\Pi }}_{c}|Q_{\Lambda }^{m-1}|{\tilde{\rho }}_{0})). \end{array}$$
where ${\tilde{\rho }}_{0}:=\Lambda \left({\hat{\rho }}_{0}\right),{\hat{\rho }}_{0}$ is the input state with state preparation noise. Equation (10) holds by Lemma 1; Equations (13) and (14) follows from Equations (6) and (Equation 4) respectively.By the definition of *Q*, we have $Q_{i,j}=\mathrm{Tr}\left(\Pi_{i}\Lambda \left({\tilde{\Pi }}_{j}\right)\right).$ Moreover, for every j it holds that
$$\sum_{i}Q_{i,j}=\mathrm{Tr}\left(\Lambda ({\tilde{\Pi }}_{j})\right)=\mathrm{Tr}\left({\tilde{\Pi }}_{j}\right)=1$$
since Λ preserves the trace. This indicates that *Q* is a Markov chain transition matrix. By the definitions of $L_{\mathrm{ave}}$ and $S_{\mathrm{ave}}$ in Equation (2), we have
(15)$$\begin{array}{c} L_{\mathrm{ave}}=\mathrm{Tr}\left(\Pi_{l}\Lambda \left({\tilde{\Pi }}_{c}\right)\right)=((\Pi_{l}\left|Q\right|{\tilde{\Pi }}_{c}))=\sum_{i\neq c^{n}}((\Pi_{i}\left|Q\right|{\tilde{\Pi }}_{c}))=1-Q_{c^{n},c^{n}} \end{array}$$
and
(16)$$\begin{array}{cc} S_{\mathrm{ave}} & =\mathrm{Tr}\left(\Pi_{c}\Lambda \left({\tilde{\Pi }}_{l}\right)\right) \end{array}$$
(17)$$\begin{array}{cc} \; & =\sum_{i\neq c^{n}}\mathrm{Tr}\left(\Pi_{c}\Lambda \left(\frac{\dim(H_{i})}{d_{l}}{\tilde{\Pi }}_{i}\right)\right) \end{array}$$
(18)$$\begin{array}{cc} \; & =\frac{1}{3^{n}-2^{n}}\sum_{i\neq c^{n}}\dim\left(H_{i}\right)Q_{c^{n},i}. \end{array}$$□

Theorem 1 demonstrates that Pauli-twirled quantum channels with leakage can be represented as Markov chains operating on distinct leakage subspaces, including the computational subspace itself. The leakage properties can be inferred from the spectral characteristics of the transition matrix, akin to analyses of RB protocols in the computational space [36]. However, this framework does not directly provide an easily applicable LRB scheme, as the transition matrix *Q* typically has a dimension of $2^{n}$, resulting in complex matrix exponential decay behavior as the number of qubits increases.

Nonetheless, estimating the leakage rate can be significantly simplified in scenarios where the number of qubits is small enough to allow manageable matrix exponential decay or when additional assumptions can be made about the leakage behavior. In the subsequent sections, we propose several physically relevant leakage noise models with straightforward theoretical exponential decay curves suitable for experimental implementation.

## 4. Average Leakage Rate for Specific Noise

In this section, we present two specific leakage noise models—single-site leakage damping noise and crosstalk-free leakage noise. We also provide the respective average leakage rates for each model.

In the following, we investigate the average leakage rate for specific leakage noise where leakage only happens on a single site (qubit). For any 1≤i≤n, we define
$$B_{i}=\left\{a\in {\left\{0,1,2\right\}}^{n}|a_{i}=2;a_{j}\in \left\{0,1\right\},\forall j\neq i\right\}$$
such that ${\{|k\rangle \}}_{k\in B_{i}}$ forms a basis of the specific leakage subspace $H_{c^{i-1}lc^{n-i}}$ where only the qubit *i* is leaking. Let $H_{l,(1)}:={⨁}_{i}H_{c^{i-1}lc^{n-i}}$ be the leakage subspace that exactly one qubit is leaking, with the corresponding basis set $B:={⋃}_{i}B_{i}$. We propose a *single-site leakage damping noise model* as a generalization to the amplitude damping noise [30]:

**Definition** **1.***Let set $W:=\left({\left\{0,1\right\}}^{n},B\right)\cup \left(B,{\left\{0,1\right\}}^{n}\right)\cup {\left\{(B_{i},B_{i})\right\}}_{i=1}^{n}\cup \left({\left\{0,1\right\}}^{n},{\left\{0,1\right\}}^{n}\right)$. Define the Kraus operators*(19)$$\begin{array}{c} \begin{array}{cc} \; & E_{kk^{\prime }}:=\sqrt{p_{kk^{\prime }}}|k^{\prime }\rangle \langle k|,\forall \left(k,k^{\prime }\right)\in W,\\ \; & E_{0}=\sqrt{I-\sum_{(k,k^{\prime })\in W}E_{kk^{\prime }}^{†}E_{kk^{\prime }}} \end{array} \end{array}$$*where probabilities $p_{kk^{\prime }},\sum_{k^{\prime }}p_{kk^{\prime }},\sum_{k^{\prime }}p_{k^{\prime }k}\in \left[0,1\right]$ for any $(k,k^{\prime })\in W$ with well-defined probabilities $p_{kk^{\prime }}$ and $p_{k^{\prime }k}$. The single-site leakage damping noise model is defined as a CPTP map * Λ *such that*
(20)$$\begin{array}{c} \Lambda \left(\rho \right)=E_{0}\rho E_{0}^{†}+\sum_{(k,k^{\prime })\in W}E_{kk^{\prime }}\rho E_{kk^{\prime }}^{†} \end{array}$$
*for any input state ρ. Denote the average leak and seep probabilities associated with the i-th site as*
(21)$$\begin{array}{c} p_{i}:=\frac{1}{2^{n}}\sum_{k\in {\left\{0,1\right\}}^{n},k^{\prime }\in B_{i}}p_{kk^{\prime }},\;q_{i}:=\frac{1}{2^{n}}\sum_{k\in {\left\{0,1\right\}}^{n},k^{\prime }\in B_{i}}p_{k^{\prime }k} \end{array}$$
*respectively.*

In the above definition, the parameters $p_{kk^{\prime }}$ can be understood as the probability of the state |k〉 flipped to $|k^{\prime }\rangle$ after the leakage damping noise, and $\Pi_{H\backslash H_{c}\cup H_{l,(1)}}$ in $E_{0}$ denotes that the noise model has no effect on the Hilbert space with leakage happens on more than one site. It is easy to check that $\sum_{i}E_{i}^{†}E_{i}=I$, hence Λ is a CPTP map [30] in Hilbert space H. Additionally, we introduce Equation (Equation 21) to simplify the representation, and we will find that the average leakage and seepage rates are only related to $p_{i}$ and $q_{i}$ for all of i∈[n]. The prefactor $1/2^{n}$ is added to fit the definition of “average” leakage and seepage rates in Equation (2).

### 4.1. Single-Site Leakage Noise

For the particular noise model described in Definition 1, we can simplify the average leakage rate from Theorem 1 as stated in the following theorem.

**Theorem** **2.***Let* Λ *be a single-site leakage damping channel as described in Definition 1. Let $p_{i}$ and $q_{i}$ be as defined in Equation (Equation 21), and assume that $p_{i}>0$ for all i and $q_{1}\geq \cdots \geq q_{n}$. Then after performing n-site LRB protocol, the expectation of the probability for measuring computational basis $p_{\Pi_{c}}\left(m\right)=\sum_{i=0}^{n}A_{i}\lambda_{i}^{m}$, where $A_{i}$ are real numbers, $\lambda_{0}\leq \lambda_{1}\leq \ldots \leq \lambda_{n}=1$, and $1-2q_{i}\leq \lambda_{i}\leq 1-2q_{i+1}$ for 1≤i≤n−1, $1-2q_{1}-\sum_{i}p_{i}\leq \lambda_{0}\leq \min\left(1-2q_{1},1-2q_{n}-\sum_{i}p_{i}\right)$. The average leakage and seepage rates of* Λ *are $L_{\mathrm{ave}}=\sum_{i}p_{i}$ and $S_{\mathrm{ave}}=\frac{2^{n}}{3^{n}-2^{n}}\sum_{i}q_{i}$ respectively.*

**Proof.** If the noise model is described by Definition 1, the corresponding condensed-operator representation only acts non-trivially on the n+1-dimensional subspace spanned by $\left\{|{\tilde{\Pi }}_{i}))||\left\{k|i_{k}=l\right\}|\leq 1\right\}$, as follows
(22)$$Q=\left(\begin{array}{cccc} 1-\sum_{i}p_{i} & 2q_{1} & \ldots & 2q_{n}\\ p_{1} & 1-2q_{1} & \ldots & 0\\ \vdots & \vdots & \vdots & \vdots \\ p_{n} & 0 & \ldots & 1-2q_{n} \end{array}\right),$$
where $p_{i},q_{i}$ are defined in Equation (Equation 21). This transition matrix can be illustrated in Figure 1. Equation (Equation 22) holds since $Q_{c^{i-1}lc^{n-i},c^{n}}=\mathrm{Tr}\left(\Pi_{c^{i-1}lc^{n-i}}\Lambda \left({\tilde{\Pi }}_{c^{n}}\right)\right)=p_{i}$, and similarly we can obtain other elements of *Q*.Although the spectrum of the transition matrix *Q* cannot be explicitly solved in the general case, it is possible to derive bounds on all its eigenvalues by examining its characteristic polynomial. For simplicity, we prove the theorem under a generic scenario where n≥2 and $q_{1}>q_{2}>\cdots >q_{n}$. In this case, it can be demonstrated that all eigenvalues of *Q* are distinct, making *Q* inherently diagonalizable. A detailed analysis of situations where algebraic multiplicities arise can be found in Appendix B.Denote $x_{i}:=1-2q_{i}$. Consider
(23)$$\begin{array}{cc} \det(Q-\lambda I) & =\left(1-\sum_{i=1}^{n}p_{i}-\lambda \right)\prod_{j=1}^{n}\left(1-\lambda -2q_{j}\right)-\sum_{i=1}^{n}2p_{i}q_{i}\prod_{j\in \left[n\right]\backslash \left\{i\right\}}\left(1-\lambda -2q_{j}\right) \end{array}$$
(24)$$\begin{array}{cc} \; & =\left(1-\lambda \right)\left(\prod_{j=1}^{n}\left(x_{j}-\lambda \right)-\sum_{i=1}^{n}p_{i}\prod_{j\in \left[n\right]\backslash \left\{i\right\}}\left(x_{j}-\lambda \right)\right). \end{array}$$
where [n]:={1,…,n}. Hence λ=1 is an eigenvalue of *Q*. Let
(25)$$f\left(x\right):=\prod_{i=1}^{n}\left(x_{i}-x\right)-\sum_{i=1}^{n}p_{i}\prod_{j\in \left[n\right]\backslash \left\{i\right\}}\left(x_{j}-x\right),$$
then the roots of function f(x) are meanwhile the eigenvalues of *Q*. Note that
(26)$$f\left(x_{k}\right)=-p_{k}\prod_{j\in \left[n\right]\backslash \left\{k\right\}}\left(x_{j}-x_{k}\right).$$As $q_{1}>q_{2}>\cdots >q_{n}\geq 0$ and $p_{i}>0$, we have $x_{1}<x_{2}<\cdots <x_{n}\leq 1$. It can be seen that $f(x_{i})$ and $f(x_{i+1})$ always have different signs, indicating a zero in $\left(x_{i},x_{i+1}\right)$ for all i∈[n−1]. As deg(f)=n, there is only one zero left to be determined, which is guaranteed to be real since all the other zeros are real. Let
(27)$$h\left(x\right)=\frac{f(x)}{\prod_{i\in [n]}\left(x_{i}-x\right)}=1-\sum_{i\in [n]}\frac{p_{i}}{\left(x_{i}-x\right)}.$$
When $x<x_{1}$, h(x) and f(x) have the same sign, and
$$1-\frac{\sum_{i\in [n]}p_{i}}{\left(x_{1}-x\right)}>h\left(x\right)>1-\frac{\sum_{i\in [n]}p_{i}}{\left(x_{n}-x\right)}.$$
Therefore we have
$f\left(x_{1}\right)=-p_{i}\prod_{j\in [n]\backslash \{1\}}\left(x_{j}-x_{1}\right)<0$,$h(x_{1}-\sum_{i\in [n]}p_{i})<0$,$h(x_{n}-\sum_{i\in [n]}p_{i})>0$,
indicating *f* having a zero in $\left(x_{1}-\sum_{i\in [n]}p_{i},\min\left(x_{1},x_{n}-\sum_{i\in [n]}p_{i}\right)\right)$.To summarize, we have a complete characterization of all eigenvalues $\lambda_{0}<\lambda_{1}<\cdots <\lambda_{n}$ of *Q*, namely
$\lambda_{0}\in \left(x_{1}-\sum_{i\in [n]}p_{i},\min\left(x_{1},x_{n}-\sum_{i\in [n]}p_{i}\right)\right)$,$\lambda_{i}\in \left(x_{i},x_{i+1}\right),\forall i\in \left[n-1\right]$;$\lambda_{n}$ = 1.By Theorem 1, the average leakage and seepage rates for the Pauli group with this specific noise equal $\sum_{i}p_{i}$ and $\frac{2^{n-1}}{3^{n}-2^{n}}\sum_{i}2q_{i}$ respectively. □

We assume in Theorem 2 that $p_{i}>0$ for all *i*. When $p_{i}=0$ for some *i*, the matrix *Q* might not be fully diagonalizable, requiring more complex data processing schemes. From a physical perspective, such complications can be mitigated by preparing the initial state such that the initial leakage on qubit *i* is negligible. Theorem 2 shows that when the seepage probability of all qubits are close to each other and close to leakage probability, i.e., $p_{i}\approx p_{j}\approx {\bar{p}}_{P}$ and $q_{i}\approx q_{j}\approx {\bar{q}}_{P}$ for all of i,j∈[n], then the multi-exponential decay will approximately collapse to two-exponential decay with $\lambda_{1}\approx 1-2{\bar{q}}_{P},\lambda_{0}=1-2q_{n}-\sum_{i}p_{i}\approx 1-2{\bar{q}}_{P}-n{\bar{p}}_{P}$. With the properties of the eigenstates for eigendecomposition of the transition matrix *Q*, we can further simplify the exponential curve to a single decay since the coefficient of $\lambda_{1}$ equals zero when the state preparation noise is negligible. The leakage and seepage of the *n*-qubit system can be consequently derived according to $L_{\mathrm{ave}}=\sum_{i}p_{i}\approx n{\bar{p}}_{P}$, and $S_{ave}=\frac{2^{n}}{3^{n}-2^{n}}\sum_{j}q_{j}\approx \frac{n2^{n}}{3^{n}-2^{n}}{\bar{p}}_{P}$, as shown in the following corollary.

**Corollary** **1.***Let the leakage noise* Λ *be as described in Definition 1 such that $p_{i}=p_{j}={\bar{p}}_{P}>0$ and $q_{i}=q_{j}={\bar{q}}_{P}$ for different i,j∈[n], and assume state preparation is noiseless, then after performing n-site LRB protocol, the expectation of the probability for measuring computational basis $p_{\Pi_{c}}\left(m\right)=A+B{\left(1-2{\bar{q}}_{P}-n{\bar{p}}_{P}\right)}^{m}$, where A,B are some real constants. The average leakage and seepage rates of* Λ * are $L_{\mathrm{ave}}=\sum_{i}p_{i}\approx n{\bar{p}}_{P}$, and $S_{ave}=\frac{2^{n}}{3^{n}-2^{n}}\sum_{j}q_{j}\approx \frac{n2^{n}}{3^{n}-2^{n}}{\bar{q}}_{P}$ respectively.*

The decay rate $1-2{\bar{q}}_{P}-n{\bar{p}}_{P}$ obtained from the LRB experiment does not provide sufficient information to fully determine $L_{\mathrm{ave}}$ and $S_{\mathrm{ave}}$. Rather, additional prior knowledge is required, such as the ratio of the leakage and the seepage rates. We postpone the proof of this corollary into Appendix C.

### 4.2. Cross-Talk-Free Leakage Noise

Previous studies have indicated that crosstalk in real devices can be significantly minimized [19]. In the subsequent subsection, we demonstrate that the exponential decay can be simplified under the condition that leakage noise occurs independently and locally across different qubits. We make the assumption that the local noise adheres to Definition 1 for each individual gate. It is important to note that in this context, the noise is inherently single-site, as each qubit possesses only one leakage site.

**Corollary** **2.**
*By performing the LRB circuit in n-site crosstalk free system for the Pauli group, the expectation of the output probability for the computational subspace of the k-th qubit is equal to $p_{\Pi_{c_{k}}}\left(m\right)=A+B\lambda_{k}^{m}$, and the average leakage and seepage rates*

(28)
$$\begin{array}{cc} L_{\mathrm{ave}} & =1-\prod_{k=1}^{n}\left(1-p_{k}\right), \end{array}$$


(29)
$$\begin{array}{cc} S_{\mathrm{ave}} & =\frac{2^{n}}{3^{n}-2^{n}}\prod_{k=1}^{n}\left(1-p_{k}+q_{k}\right)-\frac{2^{n}}{3^{n}-2^{n}}\prod_{k=1}^{n}\left(1-p_{k}\right) \end{array}$$

*where $\lambda_{k}=1-p_{k}-2q_{k}$, and A,B are some real numbers, $p_{k},q_{k}$ are leakage rates associated with Equation (Equation 21) in the k-th qubit.*


We postpone the proof of the corollary into Appendix D. This corollary can be obtained by restricting the noise in Theorem 2 to be the tensor product form of each local noise on a single qubit. Then if $p_{k}\approx q_{k}$ or we know the relationship between $p_{k}$ and $q_{k}$ with the analysis of the system, we can estimate L,S by fitting $p_{k}$ from $p_{\Pi_{c_{k}}}$ for all of k∈[n] independently. We note that the crosstalk-free noise is different from the noise defined in Definition 1, since only a single qubit can leak in Definition 1. By Corollary 2, the fitted curve associated with $p_{\Pi_{c}}$ will not follow a single exponential decay, since
(30)$$p_{\Pi_{c}}\left(m\right)=p_{\Pi_{c_{1}}}\left(m\right)\cdots p_{\Pi_{c_{n}}}\left(m\right)=\left(A_{1}+B_{1}\lambda_{k}^{m}\right)\cdots \left(A_{n}+B_{n}\lambda_{n}^{m}\right).$$
We can check that when *n* equals to 2, there will be 3 exponents, $\lambda_{1}=1-p_{1}-2q_{1},$$\lambda_{2}=1-p_{2}-2q_{2}$ and $\lambda_{3}=\left(1-p_{1}-2q_{1}\right)\left(1-p_{2}-2q_{2}\right)$ with average leakage rate $L_{\mathrm{ave}}$ = 1 − (1 −$p_{1}$)(1 −$p_{2}$) and seepage rate $S_{\mathrm{ave}}$ = $\frac{4}{5}$(1 −$p_{1}$ + $q_{1}$)(1 −$p_{2}$ + $q_{2}$) −$\frac{4}{5}$(1 −$p_{1}$)(1 −$p_{2}$).

## 5. Interleaved LRB Protocol for Specific Target Gates

In this section, we focus on benchmarking specific target gates. Benchmarking the leakage rate of an arbitrary target gate *T* differs from benchmarking the leakage rate of the Pauli group, as the target gate does not readily form the Pauli group. We propose an interleaved variant of the leakage for the previous interleaved randomized benchmarking protocol [10], named iLRB (interleaved leakage randomized benchmarking). We note that the target gate channel T can be any gate scheme, provided that the associated leakage noise model conforms to the form discussed in this section. The iLRB protocol is outlined as follows:(1)Sample a sequence of *m* Paulis $P_{1},\ldots ,P_{m}$ from $P_{n}$ and perform them sequentially to the noisy initial state ${\hat{\rho }}_{0}$ interleaved by target gate T to obtain ${\hat{P}}_{m}\circ \hat{T}\circ \cdots \circ {\hat{P}}_{1}\circ \hat{T}\left({\hat{\rho }}_{0}\right)$. Measure the output states and estimate $p_{\Pi_{c}}\left(P_{1},\ldots ,P_{m}\right)$ = $\mathrm{Tr}\left({\hat{\Pi }}_{c}{\hat{P}}_{m}\circ \hat{T}\circ \cdots \circ {\hat{P}}_{1}\circ \hat{T}\left({\hat{\rho }}_{0}\right)\right)$ through repeated experiments.(2)Repeat Step (1) multiple times to estimate $p_{\Pi_{c}}\left(m\right)$, the expectation of $p_{\Pi_{c}}\left(P_{1},\ldots ,P_{m}\right)$ under random choices of $P_{1},\ldots ,P_{m}$.(3)Sample a sequence of *m* Paulis $P_{1},\ldots ,P_{m}$ in $P_{n}$, and perform them sequentially to the prepared noisy initial state ${\hat{\rho }}_{0}$, i.e., ${\hat{P}}_{m}\circ \cdots \circ {\hat{P}}_{1}\left({\hat{\rho }}_{0}\right)$. Measure the output states and estimate $p_{\Pi_{c},P}\left(P_{1},\ldots ,P_{m}\right)=\mathrm{Tr}\left({\hat{\Pi }}_{c}{\hat{P}}_{m}\circ \cdots \circ {\hat{P}}_{1}\left({\hat{\rho }}_{0}\right)\right)$ through repeated experiments.(4)Repeat Step (3) multiple times to estimate $p_{\Pi_{c},P}\left(m\right)$, the expectation of $p_{\Pi_{c},P}\left(P_{1},\ldots ,P_{m}\right)$ under random choices of $P_{1},\ldots ,P_{m}$.(5)Repeat Steps (2), (4) for different *m*, and fit the exponential decay curves of $p_{\Pi_{c}}\left(m\right)$, $p_{\Pi_{c},P}\left(m\right)$ with respect of *m*.

When the leakage noise of the Pauli gates is negligible compared to that of the target gate T, we can benchmark any target gate $\hat{T}=T\circ \Lambda_{T}$ where $\Lambda_{T}$ has the same leakage noise as in Definition 1, by only performing the first two steps of the above iLRB protocol. In this case, we can directly leverage Theorem 2 to obtain the average leakage rate of $\Lambda_{T}$.

When the leakage noise of the Pauli gates is not negligible, however, steps (3) and (4) are needed to separate the target gate leakage from the Pauli gate leakage, and more assumptions on the target gate leakage noise are needed. We assume a specific case of the noise model in Definition 1, where the target gate T has the noisy implementation $\hat{T}=T\circ \Lambda_{T}=\Lambda_{T}\circ T$ and the noise $\Lambda_{T}$ is defined in Definition 2 with the same value for all of $p_{i},q_{j}$ in Equation (Equation 21). Similarly, we assume that $\hat{P}=P\circ \Lambda_{P}$ with noise channel $\Lambda_{P}$ as defined in Definition 2 also having same value for all of $p_{i},q_{j}$ in Equation (Equation 21).

**Definition** **2.***We define the* simplified single-site leakage damping noise *model as a CPTP map* Λ *such that*
(31)$$\Lambda \left(\rho \right)=E_{0}\rho E_{0}^{†}+\sum_{i=1}^{n}E_{0i}\rho E_{0i}^{†}+\sum_{i=1}^{n}E_{i0}\rho E_{i0}^{†}$$
*for any input sate ρ, where*
(32)$$\begin{array}{cc} E_{0} & =\sqrt{1-np}|u_{0}\rangle \langle u_{0}|+\sum_{i=1}^{n}\sqrt{1-p}|u_{i}\rangle \langle u_{i}|+\sum_{i\in S}|i\rangle \langle i|, \end{array}$$
(33)$$\begin{array}{cc} E_{0i} & =\sqrt{p}|u_{i}\rangle \langle u_{0}|,\;E_{i0}=\sqrt{p}|u_{0}\rangle \langle u_{i}|\;\forall i\in \left[n\right], \end{array}$$
*where $u_{i}\in B_{i},u_{0}\in {\left\{0,1\right\}}^{n},0\leq np\leq 1,S={\left\{0,1,2\right\}}^{n}\backslash \left\{u_{i}|0\leq i\leq n\right\}$ for any i, with $\bar{p}=\frac{p}{2^{n}}$.*

Definition 2 is to be regarded as a particular case of Definition 1, with at most a single leak happening between each $H_{c^{i-1}lc^{n-i}}\subseteq H_{l,(1)}$ and $H_{c}$. Such a simplified noise model has important applications such as measuring leakage for two-qubit gates on superconducting quantum chips. See more details in the next section. The requirement of the noise model for iLRB protocol can be further relaxed to more than a single leak between each $H_{c^{i-1}lc^{n-i}}$ and $H_{c}$ with the same leak probability.

With the assumption of the above noise model, the average leakage rate of target gate T can be estimated with exponential decay curves of $p_{\Pi_{c}}\left(m\right),p_{\Pi_{c},P}\left(m\right)$ obtained from iLRB protocol, as shown in the following theorem. Usually, the state preparation noise is negligible compared with gate and measurement noise, we also show that assuming state preparation is noiseless, we can further simplify the iLRB protocol to single-exponent decay curves in the following theorem.

**Theorem** **3.**
*For any n-site target gate T where its noisy implementation $\hat{T}=T\circ \Lambda_{T}=\Lambda_{T}\circ T$, $\Lambda_{T}$ and the noise of Pauli group $P_{n}$ both have the formations as in Definition 2 with noise parameter $\bar{p}$ be $\epsilon_{T},{\bar{p}}_{P}$ respectively, after performing the iLRB protocol, the expectation of the output probabilities*

(34)
$$\begin{array}{cc} \; & p_{\Pi_{c}}=A_{0}+A_{1}\lambda_{1}^{m}+A_{2}\lambda_{2}^{m} \end{array}$$


(35)
$$\begin{array}{cc} \; & p_{\Pi_{c},P}=B_{0}+B_{1}\lambda_{P1}^{m}+B_{2}\lambda_{P2}^{m} \end{array}$$

*where $\lambda_{1}=1-2\left(\epsilon_{T}+{\bar{p}}_{P}\right)+2^{n+1}{\bar{p}}_{P}\epsilon_{T},\lambda_{2}=1-\left(n+2\right)\left({\bar{p}}_{P}+\epsilon_{T}\right)+\left(n+1\right)\left(n+2\right)2^{n}{\bar{p}}_{P}\epsilon_{T},$$\lambda_{P1}=1-2{\bar{p}}_{P},\lambda_{P2}=1-\left(n+2\right){\bar{p}}_{P}$, and the average leakage and seepage rates for target gate T equal $L_{T}=n\epsilon_{T},S_{T}=\frac{2^{n}n\epsilon_{T}}{3^{n}-2^{n}}$ respectively. Assuming state preparation is noiseless, we can further simplify Equations (Equation 34) and (35) to*

(36)
$$\begin{array}{cc} \; & p_{\Pi_{c}}=A_{0}+A_{2}\lambda_{2}^{m} \end{array}$$


(37)
$$\begin{array}{cc} \; & p_{\Pi_{c},P}=B_{0}+B_{2}\lambda_{P2}^{m}. \end{array}$$



**Proof.** By Theorem 2, we have $\lambda_{P1}=1-2{\bar{p}}_{P}$, and $\lambda_{P2}=1-\left(n+2\right){\bar{p}}_{P}$. Since $T\circ \Lambda_{T}=\Lambda_{T}\circ T$, and $\Lambda_{P},\Lambda_{T}$ both have formations as in Definition 2, then
(38)$$\begin{array}{cc} p_{\Pi_{c}}\left(m\right) & =\frac{1}{|P_{n}{|}^{m}}\sum_{P_{1},\ldots ,P_{m}\in P_{n}}\mathrm{Tr}\left({\hat{\Pi }}_{c}{\hat{P}}_{m}\circ \hat{T}\circ \cdots \circ \hat{P}\circ \hat{T}\left({\hat{\rho }}_{0}\right)\right) \end{array}$$
(39)$$\begin{array}{cc} \; & =\mathrm{Tr}\left({\hat{\Pi }}_{c}{\left(\frac{1}{|P_{n}|}\sum_{P\in P_{n}}P\circ \Lambda_{P}\circ T\circ \Lambda_{T}\right)}^{m}\left({\hat{\rho }}_{0}\right)\right) \end{array}$$
(40)$$\begin{array}{cc} \; & =\mathrm{Tr}\left({\hat{\Pi }}_{c}{\left(\bar{P}\circ \Lambda_{P}\circ \Lambda_{T}\right)}^{m}\left({\hat{\rho }}_{0}\right)\right) \end{array}$$
(41)$$\begin{array}{cc} \; & =\mathrm{Tr}\left({\hat{\Pi }}_{c}{\left(\bar{P}\circ \Lambda_{P}\circ \Lambda_{T}\right)}^{m-1}\bar{P}\circ \left(\Lambda_{P}\circ \Lambda_{T}\left({\hat{\rho }}_{0}\right)\right)\right) \end{array}$$
(42)$$\begin{array}{cc} \; & =(({\hat{\Pi }}_{c}|Q_{\Lambda_{P}\circ \Lambda_{T}}^{m-1}|{\tilde{\rho }}_{0})) \end{array}$$
where $|{\tilde{\rho }}_{0}))=\Lambda_{P}\circ \Lambda_{T}\left({\hat{\rho }}_{0}\right)$. Let $\Lambda :=\Lambda_{P}\circ \Lambda_{T}$ with condensed-operator representation *Q*. Since the (i,j)-th element of *Q* is $Q_{ij}=\mathrm{Tr}\left(\Pi_{c^{i-1}lc^{n-i}}\Lambda_{P}\circ \Lambda_{T}\left({\tilde{\Pi }}_{c^{j-1}lc^{n-j}}\right)\right)$ for $i\in \left\{0,1,\ldots ,n\right\}$, then we have
(43)$$\begin{array}{cc} Q_{ij} & =2^{n+1}{\bar{p}}_{P}\epsilon_{T},\forall i\neq j\in \left[n\right] \end{array}$$
(44)$$\begin{array}{cc} Q_{0i} & =2Q_{i0}=2\left(\epsilon_{T}+{\bar{p}}_{P}\right)-\left(n+1\right)2^{n+1}{\bar{p}}_{P}\epsilon_{T},\forall i\in \left[n\right] \end{array}$$
(45)$$\begin{array}{cc} Q_{ii} & =1-\sum_{j\neq i}Q_{ji},\forall i\in \left\{0,1,\ldots ,n\right\}. \end{array}$$
We also provide the details for the representation of *Q* in Appendix E. Let λ be the eigenvalue of *Q*, with the representation of *Q* we have
(46)$$\det\left(Q-\lambda I\right)=\left(1-\lambda \right){\left(1-2\left(\epsilon_{T}+{\bar{p}}_{P}\right)+2^{n+1}{\bar{p}}_{P}\epsilon_{T}-\lambda \right)}^{n-1}\left(1-\left(n+2\right)\left({\bar{p}}_{P}+\epsilon_{T}\right)+\left(n+1\right)\left(n+2\right)2^{n}{\bar{p}}_{P}\epsilon_{T}-\lambda \right)=0,$$
which implies we have eigenvalues
(47)$$\begin{array}{c} \lambda_{1}=1-2\left(\epsilon_{T}+{\bar{p}}_{P}\right)+2^{n+1}{\bar{p}}_{P}\epsilon_{T}, \end{array}$$
and
(48)$$\begin{array}{c} \lambda_{2}=1-\left(n+2\right)\left({\bar{p}}_{P}+\epsilon_{T}\right)+\left(n+1\right)\left(n+2\right){\bar{p}}_{P}2^{n}\epsilon_{T}, \end{array}$$
and $\lambda_{3}=1$. Specifically, the multiplicity of $\lambda_{1}$ equals n−1. We postpone the proof of Equation (Equation 46) into Appendix F.The average leakage rate for *T* gate can then be determined as $L_{T}$ = $\mathrm{Tr}\left(\Pi_{l}\Lambda_{T}\left(\Pi_{c}/2^{n}\right)\right)$ = $n\epsilon_{T}$, and seepage rate $S_{T}=\mathrm{Tr}\left(\Pi_{l}\Lambda_{T}\left(\Pi_{c}/\left(3^{n}-2^{n}\right)\right)\right)=\frac{2^{n}n\epsilon_{T}}{3^{n}-2^{n}}$.The single exponential decay result of this theorem for noiseless preparation noise can be obtained from Theorem 3 and the properties of the eigenstates for the eigendecomposition of the transition matrix *Q*. We postpone the proof into Appendix G. □

By leveraging of Theorem 3, we can estimate $L_{T},S_{T}$ using the fitted $\lambda_{i}$ estimated from iLRB protocol.

## 6. Numerical Results

In this section, we carry out the numerical experiments for the average leakage rate of the multi-qubit Pauli group with the LRB protocol proposed in Section 3. Our iLRB protocol proposed in Section 5 can be applied to few-qubit cases, which is experimentally important to test the leakage and seepage of quantum gates. To support this, we show average leakage rates for iSWAP/SQiSW and CZ gates with prior noise according to the Hamiltonian of superconducting quantum devices in Appendix I.

### 6.1. Average Leakage Rate for Multi-Qubit Pauli Group

In this subsection, we numerically implement the LRB protocol introduced in Section 3 to estimate the average leakage rate of the Pauli group. We list two examples to show the robustness of our protocol.

Example 1 presents a simple noise model where the amplitude damping only happens in a pair of qubits between the set $B_{i}$ and ${\left\{0,1\right\}}^{n}$ for any i∈[n] in noise model 1. To show the robustness of our protocol, in Example 2 we give a more complex noise model that contains all of the amplitude dampings of the qubit pairs between the set $B_{i}$ and ${\left\{0,1\right\}}^{n}$ for any i∈[n], and we additionally add the amplitude damping for qubit pairs both in the same $B_{i}$ or ${\left\{0,1\right\}}^{n}$.

**Example** **1.***For an n-qubit circuit, we select a specific form of the noise* Λ *from the noise model in Definition 1. Let $f_{i},g_{i}$ be n-trit string denoting basis from computational subspace and leakage subspace respectively, and $f_{i}:=0\ldots 01_{i}1_{i-1}0\ldots 0,g_{i}:=0\ldots 02_{i}0_{i-1}\ldots 0$ when 2≤i≤n, and $f_{1}:=0\ldots 011=f_{2},g_{1}:=0\ldots 02$. We define the noise model as*
(49)$$\begin{array}{c} \Lambda \left(\rho \right)=E_{0}\rho E_{0}+\sum_{1\leq i\leq n}F_{i}\rho F_{i}^{†}+\sum_{1\leq i\leq n}G_{i}\rho G_{i}^{†} \end{array}$$
*where*
(50)$$\begin{array}{c} E_{0}=\sqrt{1-p_{1}-p_{2}}|f_{1}\rangle \langle f_{1}|+\sum_{3\leq i\leq n}\sqrt{1-p_{i}}|f_{i}\rangle \langle f_{i}|+\sum_{1\leq i\leq n}\sqrt{1-q_{i}}|g_{i}\rangle \langle g_{i}|+\sum_{i\in S}|i\rangle \langle i| \end{array}$$
*where $S={\left\{0,1,2\right\}}^{n}\backslash \left\{f_{i},g_{i}|i\in \left[n\right]\right\}$, and*
(51)$$\begin{array}{c} F_{i}=\sqrt{p_{i}}|g_{i}\rangle \langle f_{i}|,\;G_{i}=\sqrt{q_{i}}|f_{i}\rangle \langle g_{i}|,\forall i\in \left[n\right], \end{array}$$
*where $p_{i},q_{i}$ are uniformly randomly picked from $\left[2.5\times 10^{-5},3.75\times 10^{-5}\right]$ for i,j∈[n]. We take the number of qubits n=4 in the numerical experiment.*
*To demonstrate the SPAM robustness of the LRB protocol, we choose a specific form of noise in the state preparation and measurement processes. Assume the state preparation process has the depolarizing noise in $H_{c}$ and $H_{l}$. The resulting initial state can be denoted as*

(52)
$${\hat{\rho }}_{0}=\left(1-p_{c}-p_{l}\right)\rho_{0}+p_{c}\frac{\Pi_{c}}{d_{c}}+p_{l}\frac{\Pi_{l}}{d_{l}}$$

*where $\rho_{0}=|0\rangle \langle 0|$, and $p_{c},p_{l}$ are the depolarize probabilities with $p_{c}+p_{l}\leq 1$. The measurement noise is modeled as a perfect computational basis measurement followed by independent classical probabilistic transitions on each individual site. The probability transition matrix associated with site j is denoted as*

(53)
$$\Lambda_{M,j}=\left(\begin{array}{ccc} 1-\eta_{j0}-\eta_{l_{j0}} & \eta_{j1} & \eta_{s_{j0}}\\ \eta_{j0} & 1-\eta_{j1}-\eta_{l_{j1}} & \eta_{s_{j1}}\\ \eta_{l_{j0}} & \eta_{l_{j1}} & 1-\eta_{s_{j0}}-\eta_{s_{j1}} \end{array}\right),$$

*where $\eta_{j0},\eta_{j1}$ are 0-flip-to-1, and 1-flip-to-0 probabilities respectively, $\eta_{l_{ji}},\eta_{s_{ji}}$ are i-flip-to-2 and 2-flip-to-i probabilities respectively, where $i\in \left\{0,1\right\}$ for the j-th qubit.*

*We set parameters $p_{c}=p_{l}=0.0001$, and $\eta_{j0}=0.05,\eta_{j1}=0.1,\eta_{l_{j0}}=\eta_{s_{j0}}=0.0001,$$\eta_{l_{j1}}=\eta_{s_{j1}}=0.0005$ for any j∈[n]. The number of qubits n=4. We depict the probabilities of measuring outcomes in the computational subspace with the circuit size of Pauli gates as in Figure 2. Note that here we regard a Pauli gate as a series of Pauli X/Y/Z gates without interaction.*

*The theoretical leakage rate $L_{\mathrm{ave}}=\sum_{i}p_{i}=3.4\times 10^{-5}$, and seepage rate $S_{\mathrm{ave}}=\sum_{i}q_{i}=8\times 10^{-6}$. We fit the exponential decay curve with the LRB protocol proposed in Section 4. By Theorem 2 and Corollary 1 we see that if $p_{i}$ and $q_{j}$ are close to each other for all of i,j∈[n], then the probability $p_{\Pi_{c}}$ will be approximately collapse to a two-exponential decay with $\lambda_{1}\approx 1-2{\bar{p}}_{P}$, and $\lambda_{0}\approx 1-\left(n+2\right){\bar{p}}_{P}$. When the state preparation noise is small, we can approximate ${\bar{p}}_{P}$ via a single exponential decay ${\bar{p}}_{P}\left(m\right)=A+B\lambda_{0}^{m}$ for some constants A,B. We fit the experimental data to a single exponential decay curve to obtain ${\hat{\lambda }}_{0}=0.999957\pm 1.2\times 10^{-5}$. Then the average error $\bar{p}=\left(1-{\hat{\lambda }}_{0}\right)/\left(n+2\right)=\left(7.14\pm 2.00\right)\times 10^{-6}$, and thus the estimated average leakage rate ${\hat{L}}_{\mathrm{ave}}=n\bar{p}=\left(2.86\pm 0.80\right)\times 10^{-5}$ and seepage rate ${\hat{S}}_{\mathrm{ave}}=\frac{n2^{n}\bar{p}}{3^{n}-2^{n}}=\left(7.03\pm 1.97\right)\times 10^{-6}$. The estimated results are consistent with the theoretical ones within the errors of statistics, which verify the validity of the LRB protocol.*


**Example** **2.***The SPAM noise is set the same way as in Example 1. To show the robustness of the LRB protocol, we choose noise* Λ *which contains all of the flips (1) between subspace $H_{l,(1)}$ and computational subspace $H_{c}$, and (2) inside each subspace $H_{k}$ for all $k\in {\left\{c,l\right\}}^{n}$. We choose the number of qubits n=3. The noise strength $p_{ij}$ is picked uniformly and randomly from interval $10^{-3}\left[1,1+10^{-5}\right]$ for (i,j) in $\left(H_{c},H_{l,(1)}\right)\cup \left(H_{l,(1)},H_{c}\right)$ and $p_{ij}$ is picked uniformly and randomly from interval $10^{-6}\left[1,1+10^{-5}\right]$ for i and j both in $H_{k}$ and $i\neq j,k\in {\left\{c,l\right\}}^{n}$. By Theorem 2, the theoretical average leakage and seepage rates are $L_{\mathrm{ave}}=\sum_{i=1}^{n}p_{i}=1.51\times 10^{-5}$ and $S_{\mathrm{ave}}=\frac{2^{n}}{3^{n}-2^{n}}\sum_{i=1}^{n}q_{i}=6.41\times 10^{-6}$ respectively. By Corollary 1, we fit the experimental data using a single exponential decay curve and obtain $\hat{\lambda }=0.999974\pm 1.046\times 10^{-5}$. Then the average error $\bar{p}=\left(1-\hat{\lambda }\right)/\left(n+2\right)=\left(5.19\pm 2.09\right)\times 10^{-6}$, and the estimated average leakage rate ${\hat{L}}_{\mathrm{ave}}=\left(1.56\pm 0.63\right)\times 10^{-5}$ and average seepage rate ${\hat{S}}_{\mathrm{ave}}=\left(6.56\pm 2.64\right)\times 10^{-6}$. The numerical results validate the LRB protocol and demonstrate that the noise in the computational subspace does not affect the average leakage rate. We depict the probabilities of measuring outcomes in the computational subspace with the circuit size of Pauli gates as in Figure 3.*

### 6.2. Average Leakage Rate for Specific Gates

One important application of the iLRB protocol in Section 5 is measuring leakage of experimentally realized two-qubit quantum gates. Noise in real quantum gates can be very hard to characterize due to the complexity of gate schemes. For example, in the flux-tunable superconducting quantum devices, to implement a two-qubit iSWAP gate, the two qubits are brought to resonance adiabatically, left alone to evolve for some time duration, and finally detuned adiabatically back to their normal working frequencies [31]. Both the adiabatic evolution and the resonant evolution might lead to leakage and seepage. If one carries out the iLRB protocol for some specific gates proposed in Section 5, one would theoretically obtain one decay curve that consists of multiple exponents. A general multi-exponential decay curve is hard to fit due to statistical errors in real quantum experiments. To simplify the problem, we focus on the leakage damping noise models given in Definition 2 (The explicit form of the two-qubit case is given below). It can make the data fitting and processing more manageable. These simplified noise models are supported by the Hamiltonian evolution of the target two-qubit gates.

#### 6.2.1. Average Leakage Rate Analysis

The leakage damping noise model for iSWAP/SQiSW gate is shown below. This noise model is supported by qubits’ Hamiltonian evolution. See more details in Appendix I.
(54)$$\begin{array}{c} \begin{array}{cc} \Lambda_{\mathrm{iSWAP}}\left(\rho \right) & =E_{0}\rho E_{0}^{†}+\sum_{(k,k^{\prime })\in S}E_{kk^{\prime }}\rho E_{kk^{\prime }}^{†} \end{array} \end{array}$$
where $S=\left\{(02,11),(11,02),(20,11),(11,20)\right\}$, and
(55)$$\begin{array}{cc} E_{kk^{\prime }} & =\sqrt{\epsilon }|k^{\prime }\rangle \langle k|,\forall \left(k,k^{\prime }\right)\in S,\\ E_{0} & =\sqrt{I-\sum_{(k,k^{\prime })\in S}E_{kk^{\prime }}^{†}E_{kk^{\prime }}}, \end{array}$$
where ϵ∈[0,1/2]. This noise model contains one parameter ϵ that remained to be fitted by the iLRB experiment. Since this noise model belongs to the noise model in Definition 2, the average leakage rate of these gates can be formalized with Theorem 3.

Another commonly realized two-qubit gate in flux-tunable superconducting quantum devices is the CZ gate, of which leakage damping noise model reads (See Appendix I for more details)
(56)$$\begin{array}{c} \begin{array}{cc} \Lambda_{G}\left(\rho \right) & =E_{0}\rho E_{0}+\sum_{(k,k^{\prime })\in S}E_{kk^{\prime }}\rho E_{kk^{\prime }}^{†} \end{array} \end{array}$$
where $S=\left\{(02,11),(11,02),(20,11),(11,20)\right\}$ and
(57)$$\begin{array}{cc} E_{0} & =\sqrt{1-\epsilon_{1}}|02\rangle \langle 02|+\sqrt{1-\epsilon_{1}-\epsilon_{2}}|11\rangle \langle 11|+\sqrt{1-\epsilon_{2}}|20\rangle \langle 20|+\Pi_{H\backslash \left\{02,11,20\right\}},\\ E_{02,11} & =\sqrt{\epsilon_{1}}|11\rangle \langle 02|,E_{11,02}=\sqrt{\epsilon_{1}}|02\rangle \langle 11|,E_{20,11}=\sqrt{\epsilon_{2}}|11\rangle \langle 20|,E_{11,20}=\sqrt{\epsilon_{2}}|20\rangle \langle 11|. \end{array}$$
Similar to iSWAP/SQiSW gates, the noise model of the CZ gate learned from Hamiltonian evolution can be represented as noise model (Equation 56). Since usually, the noise for single-qubit gates is much lower than that of the two-qubit gates, we make the assumption that Pauli gates are noiseless. Comparing Equation (Equation 55) with Equation (Equation 57), one finds the leakage damping noise model for iSWAP/SQiSW gate can be treated as a special case with $\epsilon_{1}=\epsilon_{2}=\epsilon$. Thus for the more general leakage damping noise model in Equation (Equation 57), we provide the following corollary for the data analysis after carrying out the iLRB protocol.

**Corollary** **3.**
*For two-qubit target gate T with noisy implementation $\hat{T}=T\circ \Lambda_{G}$, where $\Lambda_{G}$ has the form defined in Equation (Equation 56), and we assume the Pauli gates are noiseless. Then by performing the iLRB protocol, the expectation of the output probability is $E\left[p_{\Pi_{c}}\right]=A+B_{1}\lambda_{1}^{m}+B_{2}\lambda_{2}^{m}$, where $\lambda_{i}\in \left\{1-\frac{3}{8}\epsilon_{1}-\frac{3}{8}\epsilon_{2}\pm \frac{1}{8}\sqrt{9\epsilon_{1}^{2}-14\epsilon_{1}\epsilon_{2}+9\epsilon_{2}^{2}}\right\}$, and the average leakage rates $L=\frac{\epsilon_{1}+\epsilon_{2}}{4},$ and $S=\frac{\epsilon_{1}+\epsilon_{2}}{5}$.*


We postpone the proof of this corollary in Appendix J. Here we only need the assumption that the noise of *T* gate is right hand side of T, since $E\left[P_{j}\circ T\circ \Lambda_{T}\right]=E\left[P_{j}\circ \Lambda_{T}\right]$. By Corollary 3, we can obtain the average leakage rates by fitting $\lambda_{1},\lambda_{2}$ from the exponential curve $p_{\Pi_{c}}$.

#### 6.2.2. Numerical Results for iSWAP Leakage Rate Estimation

Here we numerically analyze the average leakage rate for any two-qubit gates with leakage noise model $\Lambda_{\mathrm{iSWAP}}$. To demonstrate the SPAM robustness of the iLRB protocol, we implement measurement noise which has the same setting as in Section 6.1. Here we choose a smaller preparation noise with $p_{c}=p_{l}=10^{-6}$ in Equation (Equation 52). The leakage noise of the Pauli gate is chosen as ${\bar{p}}_{P}=5\times 10^{-6}$. The noise rate of the target gate is chosen as ${\bar{\epsilon }}_{\mathrm{iSWAP}}=\epsilon_{\mathrm{iSWAP}}/4=5\times 10^{-5}$. Hence $L_{\mathrm{iSWAP}}=2{\bar{\epsilon }}_{\mathrm{iSWAP}}=10^{-4},$ $S_{\mathrm{iSWAP}}=\frac{8}{5}{\bar{\epsilon }}_{\mathrm{iSWAP}}=8\times 10^{-5}$. By Theorem 3, we have the theoretical average leakage and seepage rates equal
$$\begin{array}{c} L_{\mathrm{iSWAP}}=\frac{\lambda_{P}-\lambda }{2(3\lambda_{P}-2)},S_{\mathrm{iSWAP}}=\frac{2(\lambda_{P}-\lambda )}{5(3\lambda_{P}-2)}. \end{array}$$

Figure 4 gives the fitted curve from simulated experimental results. From the figure, we can fit the exponent λ=0.999782(2) and pauli noise $\lambda_{P}\approx 0.999980\left(1\right)$. Hence the estimated average leakage and seepage rates are ${\hat{L}}_{\mathrm{iSWAP}}=9.9\left(2\right)\times 10^{-5}$ and ${\hat{S}}_{\mathrm{iSWAP}}=7.9\left(2\right)\times 10^{-5}$ respectively, which verifies theoretical values.

## 7. Discussion

In this paper, we proposed a framework of *leakage randomized benchmarking* that addresses the limitations of previous proposals and is more versatile in its applicability to a wider range of gates. The LRB protocol is particularly suitable for multi-qubit scenarios in the presence of SPAM noise. We presented an interleaved variant of the LRB protocol (iLRB) and conducted a thorough analysis of the leakage and seepage rates under various noise models, with a focus on the leakage-damping noise model and two-qubit gates in superconducting quantum devices. We carried out numerical experiments and see a good agreement between the theoretical leakage/seepage rates and the numerical ones for multiple gates. As the iLRB protocol is sensitive only to leakage, rather than the specific logic gate in computational subspace, it can be easily extended to other two-qubit gates realized in experiments. We leave the experimental demonstration of the iLRB protocol for future work.

One major difference between LRB and RB protocols is that single-exponential decays are guaranteed under general assumptions for RB protocols if the computational space is sufficiently twirled. However, leakage subspaces are hardly affected by any gate schemes designed on purpose for the computational subspaces, causing LRB to exhibit much more complicated decay behavior. Alternatively, gates that can twirl the leakage subspace might lead to cleaner decay behavior, but would pose somewhat unrealistic assumptions on the gate implementation that might not be experiment-friendly. In our work, we choose not to pose assumptions about the gates themselves, but instead require prior knowledge of the leakage noise models. Such prior knowledge facilitates data processing and interpretation, but their validity needs to be established either experimentally or through first-principle error analysis. Although we have proposed two simplifications under which the LRB behaviors are better understood, a more case-by-case study might be needed for other physically oriented noise models. As a complement, in Appendix J we also analyze the leakage information we can gain in a more complex noise model.

We have posed several intriguing open questions for future exploration:(1)Could we apply the LRB protocol discussed in Section 3 to compute the average leakage rate for the Pauli group, considering other Markovian and gate-independent, time-independent noise types aside from leakage damping noise?Looking ahead, is it feasible to extend our protocol to address non-Markovian noise within the entire space?(2)Is the requirement for the noise channel and gate operation to commute essential when benchmarking any gate?(3)The iLRB protocol aims to evaluate the leakage rate of the iSWAP/SQiSW gate. Experimental verification is anticipated as the next step in future work.

## Figures and Tables

**Figure 1 entropy-26-00071-f001:**
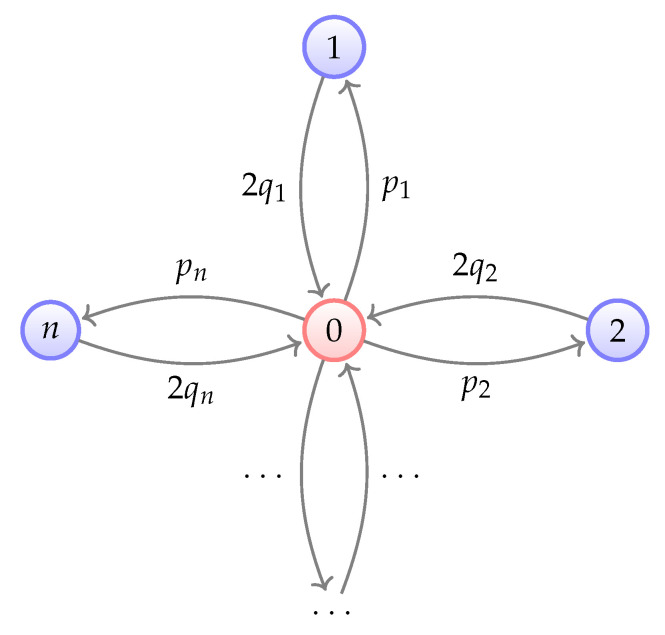
Single-site leakage model described as a Markov chain. Self-loops are omitted. Here 0 denotes computational subspace $H_{c}$, *i* where 1≤i≤n denotes the subspace where only the *i*-th qubit is leaked, i.e., $H_{c^{i-1}lc^{n-i}}$.

**Figure 2 entropy-26-00071-f002:**
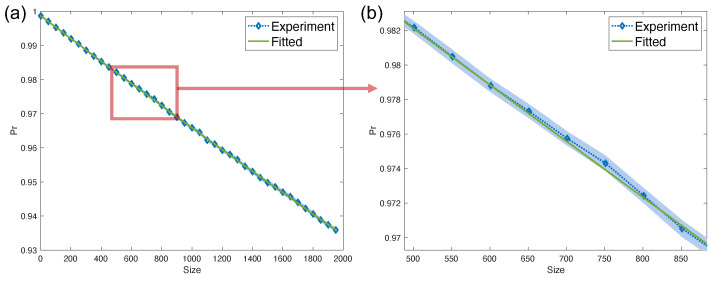
The probabilities of measuring outcomes in computational subspace with circuit size *m*. Here the probability is estimated over 200 randomly selected circuits. The vertical axis denotes the estimation for $p_{\Pi_{c}}$, and the horizontal axis denotes the size of Pauli gates sampled from *n*-qubit Pauli group. (**b**) is the zoom-in figure of the red box curves of (**a**).

**Figure 3 entropy-26-00071-f003:**
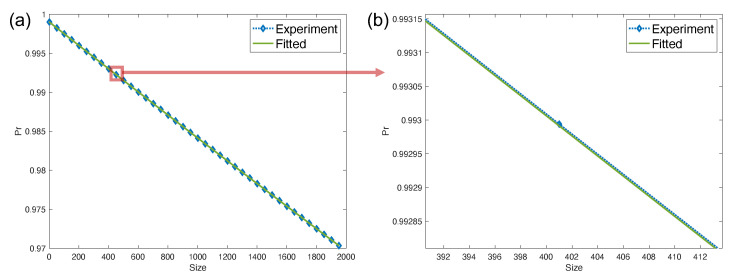
The probabilities of measuring outcomes in computational subspace with circuit size *m*. The estimation is over 200 randomly selected circuits. The vertical axis denotes the estimation for $p_{\Pi_{c}}$, and the horizontal axis denotes the Pauli gates sampled from *n*-qubit Pauli group. (**b**) is the zoom-in figure of the red box curves of (**a**).

**Figure 4 entropy-26-00071-f004:**
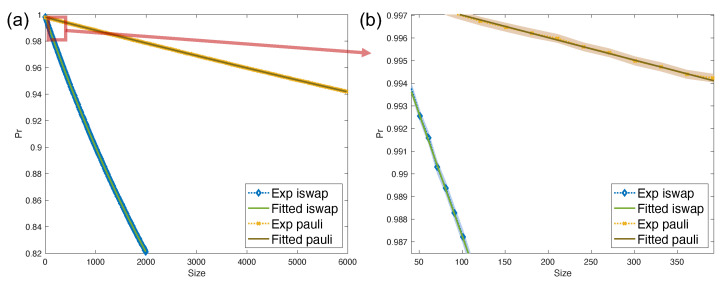
The probabilities of measuring computational subspace with the number of circuit size of iLRB protocol. (**b**) is the zoom-in view of (**a**) with an error band. Here the probability is estimated over 500 randomly selected circuits. The vertical axes for (**a**,**b**) denote the estimation for $p_{\Pi_{c}},p_{\Pi_{c},P}$ respectively, and horizontal axes denote the size of Pauli gates sampled from *n*-qubit Pauli group. (**b**) is the zoom-in figure of the red box curves of (**a**).

## Data Availability

The data presented in this study are available on request from the corresponding author.

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
