# Peer review of "Leakage Benchmarking for Universal Gate Sets"

_entropy, 2024, doi:10.3390/e26010071_

Round 1
Reviewer 1 Report
Comments and Suggestions for Authors
The authors investigate a framework named leakage randomized benchmarking for multi-qubit quantum systems. Compared with previous protocols, their scheme is not sensitive to state preparation and measurement noise. With some noise assumptions, they extend the LRB protocol to an interleaved variant called interleaved LRB. The numerical experiments demonstrate the feasibility of the two protocols. The work may be innovative and significant. However, I have the following comments,
(1) Why did the authors choose the Markovian noise model? If the system Hamiltonian does not commute with the noise operators, what kind of strategies should be used in that case?
(2) What are the resources associated with implementing this protocol? Can you use a physical model to demonstrate the corresponding noise?
(3) Superconducting qubits are utilized to implement various quantum phase gates. Could the authors demonstrate how to implement the multi-qubit gate? In real systems, various noises can affect the circuit. If there are different random noises present, what is the impact of the unknown noise?
(4) Do they introduce new noise into the process by performing repeated operations?
(5) It would be beneficial for a broader audience if the authors provide additional numerical details to enhance the comprehension of the model or method employed in the experiment.

The paper is written clearly and with good English.
Reviewer 2 Report
Comments and Suggestions for Authors
In the manuscript entitled “leakage benchmarking for universal gate sets”, Wu et al., propose a new randomized-benchmarking (RB) type protocol to characterize the leakage error for the quantum memory or quantum gates. Comparing to the former construction in Ref. [22], the proposed new scheme does not require a complete twirling on the leakage subspaces {Pi_l} and is more suitable for the multi-qubit circuit characterization. Under the noise model assumption in Definition 1, the noises under the Pauli twirling can be modeled as a classical Markovian process between different invariant subspaces, which can then be estimated similarly as the Pauli fidelities in the cycle-RB schemes. The authors then generalize the scheme to characterize the leakage error for the multi-qubit gates based on an interleaved version.
This new leakage benchmarking protocol improves the former leakage benchmarking schemes and will be useful in the task of quantum circuit benchmarking. I think this manuscript is suitable to be published on Entropy. Below is a comment:
Comparing to Ref. [22], the major limitation of this work lies on the specific noise assumption (in Definition 1). This is a reasonable assumption in most case. Will the protocol still work if the noise is not the generalized damping noise (but still Markovian and time-independent)?
